# Monitoring Track Transition Zones in Railways

**DOI:** 10.3390/s22010076

**Published:** 2021-12-23

**Authors:** Roberto Sañudo, Ignacio Jardí, José-Conrado Martínez, Francisco-Javier Sánchez, Marina Miranda, Borja Alonso, Luigi dell’Olio, Jose-Luis Moura

**Affiliations:** 1Transport Department, School of Civil Engineering, Universidad de Cantabria, 39005 Santander, Spain; borja.alonso@unican.es (B.A.); luigi.dellolio@unican.es (L.d.); joseluis.moura@unican.es (J.-L.M.); 2Ferrovial, 28042 Madrid, Spain; ignacio.jardi@ferrovial.com; 3ADIF—Administrador de Infraestructuras Ferroviarias, 28045 Madrid, Spain; JCMARTINEZ@ADIF.ES; 4PRECON S.A.—Precon Soluciones Prefabricadas, 28003 Madrid, Spain; fsanchez@precon.cemolins.es; 5Geotechnics Department, School of Civil Engineering, Universidad de Cantabria, 39005 Santander, Spain; marina.miranda@unican.es

**Keywords:** track transition, monitoring, transient analysis, vertical displacements, shear stress, vertical accelerations

## Abstract

This manuscript presents the first measurement program and data collection on the Dinatrans track transition solution after it was installed in a track section in the north of Spain (Galicia). The Dinatrans solution was created to address the limitations of several track transition solutions. This novel solution consists of two inner and outer rails from slab track to ballast track, pads with different stiffness over sleepers of variable lengths installed from ballast track to slab track, and a simple substructure formed by non-structural concrete poured over the natural ground. The main objective of this research was to assess the suitability and the initial performance of the Dinatrans track transition solution. The measured variables for these initial real-world tests were vertical accelerations on sleepers, shear stress on rails, vertical displacements on rails and vertical displacements on sleepers. All measurements of these variables were obtained in an in-situ program by installing vertical accelerometers and LVDTs on the track structure and extensometer gauges on the rails and sleepers. The methodology and the procedures followed are described. The Dinatrans initial solution was compared with the Standard solution used in Spain using these initial measurements. This field analysis provides an initial understanding of the performance of the new track transition. Further measurements will be required to check the track transition performance over the long term; however, no maintenance works have been necessary since construction (2016).

## 1. Introduction

Track transition zones in railways are the solutions which are applied to try to improve problematic issues that occur when there is a sudden change in track stiffness [1,2,3]. Many problems occur in these track areas, not only in the track’s structure but also in rolling stock where passengers can suddenly notice when they are travelling over these track areas [2]. The abrupt change in stiffness and differential vertical deflections between each side of the transition can be detrimental to track structure, cause accelerated wear, material loss, and hanging sleepers, among other problems.

Descriptions of the problem and modeling methods, along with many solutions, can be found in the literature (some of which have been extensively analyzed and studied) [2,3]. This study provides a new analysis of these special track points. It describes the behavior of a new track transition solution by comparing the track measurements obtained from two track transition solutions: the “Standard” solution in Spain applied by the main railways administrator (ADIF) and the new solution, Dinatrans [4].

Many solutions to the problems resulting from different track stiffness and differential vertical deflection occurring in these areas have already been proposed in [1,2,3]. The solutions proposed are classified according to four different types in Table 1.

Both solutions evaluated in this research are mixed solutions as they combine different elements. In the same order as followed by the type of action taken, the mixed solutions can be categorized as either combining elements from the same solution, e.g., 4.1, 4.2, and 4.3 or combining solutions e.g., 4.4, 4.5 and 4.6.

According to these subcategories, the Standard solution and the Dinatrans solution can be found in subgroup 4.4. These “mixed solutions” are new solutions which use similar materials and use the same type of approach as previous solutions. This means that “mixed solutions” are based on previous solutions but are more complex.

There is a vast literature about track transitions, problems and solutions but there are few references in relation to track transition monitoring. Monitoring experiences with solutions involving track transitions in backfills using bound and unbound granular materials can be found in [5] and solutions with under sleeper pads in [6].

In relation to monitoring and measuring techniques at track transitions, the literature review offers some important insights. For example, [7] suggests two categories when analyzing vertical displacements; permanent and transient, depending on whether they involve use of convenient equipment applied previously or the application of digital image correlation (DIC-based) devices in railways to measure the dynamic displacement of rail along the track.

In relation to track transition monitoring techniques, [5] describes a complete monitored wedge-shaped backfill for transition zones. The usual variables to analyze track performance in these areas were obtained including shear deformations of the rail web using strain gauges, fiber-optic sensors to assess seat loads on the rail pads, rail vertical displacements using contactless systems (LASER + position sensitive detector (PSD)), rail-sleeper vertical relative displacements using LVDT, and vertical acceleration of the sleeper using micro-electro-mechanical systems (MEMS) and piezoelectric accelerometers. Researchers in [8,9,10] focused on the instrumentation and performance monitoring of railroad bridge approaches with multi-depth deflectometers. Track settlement data acquired over time are presented to compare the contributions of different substructure layers with the permanent deformation accumulation. Some studies have focused on fasteners and show how to control these elements in track transitions [11] while others have suggested the use of satellite radar data to monitor track performance in track transitions [12]. A complete study of track transitions, considering measurement, assessment and means of improving them can be found in [13].

Track transition effects due to abrupt changes in stiffness can cause accelerated wear on the track. This sudden difference in stiffness in the transition zones (from ballast to slab or vice versa) can increase the interaction forces for all frequencies up to 150 Hz [14]. Train speed can increase this effect and multiple loads can induce additional oscillatory behavior that is not typical of single loads [15].

Track transition zones, from embankment sections to bridge sections (or the reverse), can suffer external forces (not only from trains but other sources), for example, seismic motions, and differential settlements during the seismic response. Researchers in [16], analyzed this special case and proposed additional countermeasures, such as anti-buckling plates and ballast-retaining walls to prevent rail buckling.

During the last decade, all the elements from both track transitions have been calculated, tested and are being assessed. Recent investigations of track transition with additional rails, where results have shown a reduction of rail deflections, track accelerations and rail pad forces are described in [17]. In other studies, it has been suggested that fastening systems based on pad stiffness, baseplate weight and the arrangement of baseplate fastening systems, should be carefully applied to maximize the benefit of baseplates at transition zones [18]. The results of using gradually decreasing pad stiffnesses over the larger slab track area provided a better smoothing effect on the transition zone. Under sleeper pads (USP), as supporting elastic elements, are being used to decrease stress and vibration on the ballast; the integration of the USP component into the track has been shown to provide an approximately 25% decrease in ballast acceleration [19]. The main justification for installing USPs at transition zones is to reduce sleeper-ballast contact forces, aiming at reducing ballast degradation. These USPs can control the vertical stiffness of the track, and reduce the loads transferred by the sleepers to the ballast layer [20]. The stiffness of track transitions can also be modified by installing rubber mats between track layers [21].

The condition of track transition materials, for example, ballast moisture (e.g., ballast and sub-ballast), can also influence track transition performance. Dynamic loads increase with ballast moisture and this moisture condition is another source of degradation in these areas [22]. Another example of this is the influence of the dynamic elastic modulus of the graded broken stone on the dynamic response of the vehicle system which is larger than that of the dynamic elastic modulus of the subgrade bed surface layer [23].

In light of these previous studies the authors decided to monitor important and relevant variables in track transitions, such as vertical acceleration on sleepers, vertical displacements and shear stresses in rails. The special case considered in this analysis presents some differences with respect to the described cases. During the monitoring program, the track section was not yet open to traffic (despite this, it was necessary to estimate the initial behavior before opening), the maximum train speed was 80 km/h, and track transition was from ballast to slab (open track-tunnel).

The experience in Lugo allowed the authors to compare two different solutions from the outset, considering the Dinatrans solution and numerical model and the influence train travel direction has on track transition performance.

The article is structured as follows. Following the introduction, the next section presents the case study, with a brief description of the solutions, experimental measurements and the transition comparison. Section 3 explains the Dinatrans numerical model. Section 4 describes the limitations of the study and the statistical analysis used to compare both solutions (it was necessary when the results were so similar). The manuscript closes by outlining the main conclusions drawn, along with suggestions for further work.

## 2. The Case Study

The following section presents and describes both track transition zones analyzed, the commonly used and the Dinatrans solution.

### 2.1. Track Transition Description

The Dinatrans design is aimed at creating an improved track transition solution. The initial development was based on 2D dynamic simulations (explained in Section 3). The initial models developed for the Dinatrans solution and an in-depth description can be found in [4,24,25,26,27], respectively.

The track section chosen was located in a commuter network close to Puebla de San Julian (in the province of Lugo, Spain). This section is part of the high-speed Ourense-Monforte-Lugo (branch of Puebla de San Julian) line. It was built in 2016 and has been in service since April 2018. Several tunnels are located along the network. Tunnel number 3, where the transition zone was built, is 1344.3 m long, between kilometric points 3+483.60 and 4+827.90, and uses a BPP14-type slab-track superstructure. The high-speed ballast track section consists of UIC 60 rails, fastenings (k = 100 kN/mm), polyvalent sleeper PR 01-C60 and a formation layer over a granite sub-base. The tunnel (n°3) slab track consists of fastening (K = 33 kN/mm) polyvalent biblock sleeper BPP14 embedded in HA 35 concrete over an HA30 concrete sub-base.

The Standard track transition is located at PK 4+827.90 (outside the tunnel) (Figure 1a) [2,4] and consists of the following main elements:

(1) Additional inner rails: two 15.5 m long inner rails placed over 10 sleepers on the slab-track side and over 15 sleepers in the ballast side; (2) The pads under the rail are of variable stiffness: 33 KN/mm, 60 KN/mm and 80 KN/mm; (3) 15 monoblock concrete sleepers close to the slab with pads of 60 KN/mm and under sleeper pads (USP) of 200 KN/mm, PR0I sleepers; (4) A reinforced concrete beam is located under the ballast (HA-35) close to the 10.30 m slab joined trough connectors; (5) A small concrete wall protects the ballast at one side of the transition.

The Dinatrans solution is located at PK 3+483.60 (Figure 1b) [4,24]. This new solution consists of:

(1) Additional rails, (a) two 18 m long outer rails (UIC60) placed over 20 sleepers on the ballast side and the concrete slab on the slab track side, (b) two 22.8 m long inner rails (UIC60) over 9 sleepers on the slab track side and over 28 sleepers on the ballast side; (2) Different pad stiffness was used to provide continuity in track stiffness. For the first 12 sleepers close to the slab track side with pads of 60 KN/mm were used, the next 12 sleepers used 80 KN/mm pads and the last 16 sleepers close to open ballast track section used 100 KN/mm pads; (3) Longer sleepers. Five groups of 8 sleepers each having different lengths of 4 m, 3.68 m, 3.36 m, 3.04 m and 2.72 m running from the slab track to the ballast track. Under sleeper pads were not used; and (4) A constant 30 cm thick layer of concrete under the ballast (HM-25). More detailed information about both solutions can be found in previous studies [2,4,24].

One common solution to make a gradual track transition in stiffness is by changing the stiffness of the pads gradually. It is practical and it is easy to install. Therefore, the authors sought to retain it for the Dinatrans solution. For the analyzed railway track (ballast side has 100 KN/mm pads), the slab track used (BP-14) is similar to the Rheda system (with pads of 33 KN/mm). It is necessary to pass from 33 KN/mm to 100 KN/mm which is why pads from 60 to 80 KN/mm were used to make the transition smooth.

In order to attenuate and mitigate longitudinal track vibration from slab to ballast and avoid ballast decompaction in the vicinity of the slab, the Dinatrans solution uses an under ballast mat cross section panel between slab and ballast (Figure 2).

The length of the rails was selected to create a gradual change in vertical displacements and stress in the ballast. The ballast thickness is constant, so variation in track stiffness is due to rails with variable length, variable sleeper length, and variable pad stiffness. To achieve this, six configurations of sleepers (with variable length), and six different configuration of rails, inner and outer rails were previously numerically analyzed in [4,26,27,28]. The best solution was the one with length of inner rails 22.8 m and outer rails 18 m. This solution makes the track transition in vertical settlements and vertical stresses more gradual.

In this numerical analysis the rail section used in calculations was UIC 60; the remainder of the outer and inner rails used the same section.

All the elements of the rails (main rails, inner rails and outer rails) to ballast were simulated to provide a gradual vertical displacement and vertical stress in ballast along the track transition.

A transient analysis was carried out. The study involved the following activities:

(1) checking the initial transient values for all the variables to make sure they were within the allowed values. Regarding structural integrity, (2) comparing the transient values at the beginning of selected variables in both track transitions, (3) assessing how the model compared with the real solution and (4) assessing how the train’s direction of travel affected the performance of the track transition.

Both track structures were placed over a good quality terrain, consisting of granite rock with good geotechnical properties. The next table shows the most important geotechnical characteristics from the in-site tests.

Table 2 shows the average values taken from the in-situ test undertaken during the project phase. Field surveys showed granite rock (good quality and fractured) along the analyzed section. The variables were: E (MPa) Young’s modulus of the material (average), u Poisson’s coefficient, ρ (t/m^3^) density of the natural ground, C (MPa) cohesion of the material and the dip j (°).

### 2.2. Experimental Measurements

To compare the initial performance of the two solutions a track measurement program was designed. Several track variables needed to be measured to provide a basis for making comparisons. The authors decided to focus on vertical displacements in rails and sleepers, vertical accelerations on sleepers, and stress in ballast. These variables were selected based on previous research and a state-of-the-art analysis [4,25,26,27,28]. When the field measurement program was designed, stress in ballast was difficult to obtain so the authors decided to measure shear stress in rails. As a result of the previous state-of-the-art analysis in track transition performance, the following variables for the transient analysis and comparison at the initial stage were chosen: (1) vertical accelerations on the sleepers (vertical accelerometers), (2) shear forces in rail web (extensometer gauges), (3) relative vertical displacements (potentiometers), and (4) vertical displacements of the sleeper (LVDTs). This is shown in the following figures. Figure 3 shows the instrumentation tools (e.g., LVDTs, potentiometers, extensometer gauges and vertical accelerometers). Figure 4 shows the position of these devices on the sleepers and rails (in the different sections). Table 3 shows the characteristics of all sensors used during the field tests.

Figure 5 shows the sensor position during field measurements in the Dinatrans solution. In order to compare the performance between both tracks’ transitions, the instrumentation in the standard solution was placed in the same location (symmetrically when looking from the slab to the blast area). Both track transitions were divided into eight sections. Two first sections were slab track in the tunnel structure (BP-14 type) and the remaining 6 sections were in the ballast track (at both sides of the tunnel), as shown in Figure 5. Figure 5 was an open ballast track. The sleepers and supports were numbered from the limit between slab track and ballast track to both sides. The slab-track side had thirteen sleepers (from 1 to 13) and the ballast side (in ballast track) had forty-three sleepers (from 1 to 43) (Figure 5). The position of the devices along the track transitions was chosen to assess the overall performance of the track transitions during the passing of trains. Devices were placed on the slab-track, the ballast track, at the end of both track transitions, and some sections in the track. The measurement points and devices are summarized in Table 4. For example, section 4 in the ballasted track had one monitored sleeper (number 11) and section seven in the ballasted track had one monitored sleeper (sleeper 35).

The moving material used for the test was a track works locomotive (Figure 6). The track works locomotive with 1364 HP of power weighed 83.4 tons and had two motorized bogies. The overall length was 16.237 m and maximum speed was fixed at 80 km/h which was divided into steps of 20 km/h having train velocities of 20, 40, 60 and 80 km/h. The trains travelled in both directions. Odd number passes corresponded to the train running form slab track to ballasted track and even number passes were when the train travelled in the opposite direction from ballasted track to slab track. The measurements were taken before this section was opened for the first time (Table 5), during the first week of July 2016. A total of 26 trains passing per transition solution were measured, 6 to 8 passes per each train velocity to assess the influence of the number of passes. The sensors were placed during the measuring program only, and then they were dismantled. There was not a continuous monitoring of the solutions, but the authors are in conversation with the administration to start a new measurement program under normal traffic conditions.

The sampling frequency chosen was 2 Khz. It is normal to use a lower sampling frequency in high speed and conventional tracks but here the researchers wanted to obtain more data. After applying filters of 500 Hz, 250 Hz and 200 Hz, and in order not to lose valuable data from the measurements, the researchers decided to work with the raw signal, which was used to obtain maximum and minimum values from both track transitions (i.e., peak values).

### 2.3. Track Transition Comparison

Measurements of the variables previously mentioned were obtained for both track transitions in both directions. The values from the figures in this section, reflect the maximum and peak values for all data acquired during the monitoring campaign. Each line represents a passing train over the track transition. Continuous lines represent values measured for the Dinatrans solution, while dashed lines are the corresponding values for the standard solution. The values represented were selected from the initial passing trains, middle trains, and the last passing trains to evaluate the overall performance for all train passes. The values between them have been omitted for a better understanding and to be close to the chosen ones. These values can be seen in [4].

Figure 7 shows the peak values of vertical acceleration in sleepers for each passing train in both track transition solutions.

Point AT21 (2), (X = 2 in Figure 7) registered vertical accelerations of under 2.94 m/s^2^, regardless of the train speed. The points with the highest values of vertical acceleration corresponded to the sleepers closest to the ballast open track section AT51(5), AT71(6) and AT81(7) with more than 58.8 m/s^2^. Figure 7 shows that the vertical acceleration values reached a constant trend with train passes. Negative values were symmetrical with the *x* axis and are, therefore, omitted. The maximum values for the first, tenth and 25th passing trains are depicted, with the remaining values falling between them. Negative values were similar to positive. In other studies, vertical accelerations in concrete sleepers were found to reach values ranging from 29.4 m/s^2^ to 107.8 m/s^2^ [29], meaning the vertical accelerations on sleepers fall within these limits. Other authors have obtained values between 17.64 m/s^2^ or 19.6 m/s^2^ in wooden sleepers [30]. In both cases field measurements were obtained in the extreme of the sleeper. Dinatrans and standard values were taken from the middle of the sleeper. Due to the fact there was a small cant, this form of measurement does not introduce problems and produces an average value across all sleepers. Previously, values (from references) can give an order of magnitude and an approach. Other values up to 80 m/s^2^ and impact values of up to 100 m/s^2^ have been observed (these were obtained from laboratory tests) [31]. Higher values of vertical accelerations in track structures are expected with higher values of stiffness [31].

In the case of vertical stress on rails (Figure 8), the maximum/minimum values recorded during each passing train determine the most unfavourable stresses supported by the rail when the locomotive travels through. As might be expected, the stress values obtained during each train pass over the common transition zone were similar at all points, both at lower and higher speeds.

The point with the highest registered values was point GC21(2) (section 2, sleeper 2 see Table 4), for all passing trains and, as in the previous case, the Dinatrans transition zone provided similar stress values at all points, both at lower and higher speeds. The shear stresses reached similar values for both track transition solutions.

Normal values for rail shear stresses for UIC 60 rails are between 72 and 92 MPa [32], or rail breakage at over (880 MPa) [33]. The measured values were found to be far from these critical values.

In the case of vertical rail displacements, the minimum value recorded for each train pass determines the highest vertical deflection of the rail with respect to the sleeper. Note that during the higher speed passes, the displacements at point PV11(1) (section S1, sleeper number 11) were around 80 km/h, making these movements less regular than in those found for the lower speed passes.

Figure 9 shows that the highest values of vertical displacements between rail and sleeper were in the slab track. This was due to vertical deflection in the pad of the BPP14 slab track system. The minimum value recorded for the vertical displacements of the sleeper for each train pass determines the highest descent of the sleeper.

In the common transition zone solution, all the vertical displacements were found to be below one millimeter at all points (Figure 9) and all the vertical displacements of the sleeper (Figure 10) had very similar values in all the passes. The same results occurred for all the vertical displacements in the Dinatrans transition zone and in both types of transition zone most of the vertical displacements of the sleeper showed very similar values during all passes. The values obtained were found to be below one millimeter at all points, except at point PT31(1) (section 3, sleeper 2) which experienced decreases ranging between 1.0 and 1.4 mm. This area (section 3) showed the highest vertical displacements. Sleepers here could possibly be hanging sleepers, which would explain the greater values in both track transitions.

The trend for Figure 7, Figure 8, Figure 9 and Figure 10 was to remain horizontal with the passing trains. Figure 8 gives an example of the trend which was similar to the rest of variables when passing trains increased. This gives an indication of the performance of the track transitions.

These initial measurements are under the allowed limits: vertical accelerations [29,30,31], shear stress in rails [32,33] and vertical displacements [34].

The data obtained from the track measurement program also allowed the authors to assess the vertical track stiffness in sleepers 11, 19 and 35 using the following formulae taken from Esveld (2001) [35].
(1)$$K_{d}=\frac{a}{4}\cdot \sqrt[{3}]{\frac{Q^{4}}{EI\cdot W_{max}^{4}}}$$
(2)$$L=\sqrt[{4}]{\frac{4\cdot EI\cdot a}{K_{d}}}$$
(3)$$K_{total}=\frac{8\cdot EI\;}{L^{3}}\;\;\;$$
where *K_d_* (N/m) is the spring constant of discrete support, *K_total_* (N/m/m) is the total spring constant of the track, *a* (m) is the sleeper displacement, *Q* (N) is the wheel load over the support, *EI* is the rail bending stiffness (N·m^2^), *W_max_* is the maximum vertical displacement of the rail (m), and *L* (m) is the characteristic track length. This information was obtained for train passes in both directions to assess whether train travel direction had a bearing on the results for vertical stiffness. Figure 11 shows the total spring constant for track *K_total_* for different sections and sleepers and for each train travel direction. The Dinatrans stiffness was higher than for the common solution and the track superstructure was also more fastened in the Dinatrans solution. The Dinatrans solution created a rigid track frame between rails and sleepers embedded in ballast which more effectively tied the railway track. The stiffness was similar in both solutions only for the first section which corresponded to slab track. It is also of note that stiffness showed little variation depending on the train travel direction where the values were found to be around 100 kN/mm and 120 kN/mm. Initially values were found to reach 180 kN/mm on the slab track side, but these values tended to stabilize with the number of passing trains and approached values closer to 120 kN/mm.

The literature shows that the optimum stiffness values to save overall costs (maintenance + dissipated energy) for normal track are in the region of 70–80 kN/mm [36], or approximately 75 kN/mm [37]. Values of between 100 kN/mm and 120 kN/mm are normal on high-speed tracks [38]. Esveld (2001) provides stiffness values of 100 kN/mm for a classic ballast track [35]. Therefore, the stiffness values obtained are in the range of expected values. On both tracks, transitions behaved in a similar way when these initial measurements were taken. Values recorded were under the normal limits in both cases. In order to check the theoretical model, the authors compared the numerical results with the actual track measurements.

## 3. Numerical Model

A good review of modeling techniques in track transitions is provided in [1,2,3] and there are recently reported techniques that consider, for example, ballast settlement [39,40]. It is necessary to use nonlinear contact elements to model the interface between sleepers and ballast so the hanging sleepers phenomenon can be better studied. To study the ballast settlement in areas with stiffness variations in railway tracks, the ballast is modeled as a linear lattice [41].

To carry out this research, a numerical 2D model similar to the ones used in [4,25,26,27,28] was used, consisting of 200 m straight track divided into elements of 0.05 m size. The Dinatrans numerical model consisted of a series of spring and damper systems for the vehicle and the track. Rail, sleepers and slab are considered inertial materials with mass, density and mechanical properties according to Young’s modulus and Poisson’s ratio. Rail pads, ballast, and formation and foundation layers are characterized by stiffness and damping properties. The geometric and mechanical properties of the materials can be found in Table 6 and Table 7, respectively. Ground material properties were taken from the field test performed during the track construction project.

A scheme of the complete model vehicle and track is shown in Figure 12 (upper) and the discretization of a track section is shown in Figure 12 (lower). The left side corresponds to slab track and the right side corresponds to ballasted track.

The train model is a combination of spring and damping elements, as presented in Figure 12. The model assumes the following: (1) the track is modeled in a straight section (no curves), (2) only vertical loads are considered, (3) there is symmetry with respect to track axle. The modeled moving train was an ALCO 313 Locomotive. The geometrical characteristics are shown in Figure 6 and the mechanical parameters are provided in Table 8.

Ballast thickness was 30 cm (see Table 6) and its density was 1.9 Tn/m^3^.

The authors sought to modify sleeper (pads) and rail lengths to improve the track performance under service by reducing vertical displacements and stress in ballast (ballast thickness kept constant).

A comparison of the theoretical 2D modelling of the Dinatrans solution with the measurements was performed. Both train travel directions, from ballast to slab and from slab to ballast were represented, with maximum values shown in the upper graphic and minimum values shown below.

The model trends were similar to real track performance. Values of shear stress (Figure 13 sleeper 2 section 1, Figure 14 sleeper 11 section 4) tended to be constant with the speed of the passing trains. Negative shear stress was higher in the real measurements than in the model estimation (i.e., around 10 MPa).

Figure 15 and Figure 16 show vertical displacements of sleepers 11 and 19 in relation to train speed. The positive values are seen to be similar in both the model and the real measurements. Vertical displacements are slightly higher in the model than in the real test. This difference could be due to the lack of vertical track stabilization caused by traffic circulation over these areas when the measurements were taken (as addressed in the limitations explained in the next section).

Figure 17 shows the vertical displacements of rail from sleeper number 35. The values correspond to the rail pad vertical deflection under the passing trains for both train travel directions. The values resulting from the test were higher than those provided by the model. There was a small difference of about 0.1 mm from the numerical model and the track transition on-site measurements. In both cases, these vertical deflections followed a similar trend and appear to have been constant with train speed for both train travel directions. This could be due to the low velocities of the train. The small differences between the model and the measurements could be attributed to the fact that the section was not opened to real traffic and only track working machines (e.g., tamping machines, rail grinding machines, dynamic track stabilizers, etc.) had passed over the analyzed section before the measurements.

## 4. Analysis and Discussion

Numerical simulations were used to compare the results from both track transitions. Some of these simulations gave some advantage to the Dinatrans solution over the common solution. The following important limitations needed to be considered during this initial phase of the experiment:

(1) The track was stabilized only with a dynamic stabilizer and no real train movements had taken place on the tracks at the time of the measurements. Only work locomotives, tamping machines and dynamic stabilizers passed through this track section before the data was obtained (the network by-pass was operational from April 2018).

(2) The maximum velocity of the locomotive was 80 km/h. This was the limit because the track site was part of a by-pass track and the train had just 3 km to operate in as it was not connected to the main line when the measurements were taken. The train required a minimum distance to brake under safety conditions. It is difficult to reach dynamic effects for these low speeds [42,43].

(3) Both track transitions were built on both sides of a slab track tunnel. Despite this attempt to achieve the same initial boundaries, the geometrical conditions of the ballast track were slightly different. The ballast and sub-ballast thickness were the same, the track cant varied between 0.9 and 1.4 cm (ballast thickness experiment small deviations around 0.3 m), concrete layers were formed from different materials, sub-base and natural ground materials were the same in both solutions.

(4) Both measurements showed low values (less than 25% difference, see Table 9) for both track transition solutions, which is normal as these are the first tests on these recently built track transitions in Spain.

Despite these limitations, the measurements were taken from the same reference points from the slab track in the tunnel for both solutions. These measurements are valid for making initial comparisons between both track transitions.

As shown by the previous results, both solutions performed similarly. Some of the values were generally found to be very similar for both solutions (less than 10% difference, see Table 9).

The values for the measured magnitudes increased with the number of passing trains but all reached a maximum which appeared to be constant. All values appeared to be similar for both track transitions but limitation number 3 gives some advantages to the common solution over the Dinatrans solution. In spite of this, the Dinatrans solution performed well.

An initial comparison between both solutions was required before the opening of the line to real traffic. It was very difficult to compare in this situation because the track had not definitely settled. Regarding signal processing, if the RMS value is obtained by digital processing of a sequence of signal samples, both the uncertainty and the bias of the measured value depend on the algorithm used [44]. In order to make initial comparisons, maximum values are better for detecting earlier problems [45]. Comparing maximum and minimum values is acceptable when values are close and the corresponding hypothesis test allows it.

It was very difficult to compare such small and similar values; however, this problem was overcome by using statistical analysis. Using the maximum values of each train pass on each support, the means were calculated for each direction for both solutions (see Table 9).

Once the means of all the maximum values were obtained, a comparison between both types of solution was carried out through an indicator that represents the improvement of one with respect to the other, through the formulation [4]:(4)$$I_{\max}=\frac{{\bar{x}}_{DMAX}}{{\bar{x}}_{EMAX}}\cdot 100$$
(5)$$I_{\min}=\frac{{\bar{x}}_{DMIN}}{{\bar{x}}_{EMIN}}\cdot 100$$
where *I*_max_ and *I*_min_ are the indicators that give the percentage improvement of the Dinatrans solution with respect to the common solution for the maximum and minimum values (see Table 10).

As the data was so dispersed, the authors decided to carry out statistical hypothesis testing to see if any differences existed between one solution and the other.

Using the maximum values for each train travel direction:

**Hypothesis** **1** **(H1).**${\bar{x}}_{DMAX}={\bar{x}}_{EMAX}$*Null hypothesis, both maximum values are equal*.

**Hypothesis** **2** **(H2).**${\bar{x}}_{DMAX}\neq {\bar{x}}_{EMAX}$*Alternative hypothesis, both maximum values for Dinatrans and common measurements are different*.

Using the minimum values for each train travel direction:

**Hypothesis** **3** **(H3).**${\bar{x}}_{DMIN}={\bar{x}}_{EMIN}$*Null hypothesis, both minimum values are equal*.

**Hypothesis** **4** **(H4).**${\bar{x}}_{DMIN}\neq {\bar{x}}_{EMIN}$*Alternative hypothesis, both minimum values for Dinatrans and common measurements are different*.

The null hypothesis assumed that the means were equal. The alternative hypothesis was that they were not the same. In our case, we wanted the null hypothesis not to be fulfilled so that the results are comparable.

The results from the contrast test are presented in Table 10 (Student’s *t*-test values). By looking at this table, all those values in which the Students-t statistic is close to or greater than 1.96 (in absolute value) are significantly different at the 95% level and therefore it makes sense to carry out this comparison. If the t-statistic values are lower than that value (in absolute value), it means that there are no significant differences between one solution and the other.

Table 10 shows the values of the indicators obtained and the t-statistic values. The Student’s t statistics are displayed to see which values can and cannot be meaningfully compared. The underlined values are not sufficiently statistically significant to be considered in the comparisons (such as vertical acceleration and δ_sleep_) (Table 10).

In general, the acceleration values were not significantly different for both directions of movement, so meaningful comparisons cannot be made, the values were very approximate. The measured stresses show that the values of the statistics are higher than 1.96 (in absolute terms), indicating that there are significant differences, which means they can be compared. However, the values of the t statistic for the vertical rail displacements are low, so there is no significant difference to establish comparisons. Meanwhile, the vertical sleeper displacements (δ_sleep_) of most of the values can be compared because the differences are significant.

The vertical accelerations when going from slab track to ballast track were not significantly different (the values are very small and close, Table 10). However, there was a significant difference in the minimum vertical accelerations when the train goes from ballast track to slab track. The minimum vertical accelerations when the train traveled from ballast track to slab track were 22.55% higher in the Dinatrans solution (Table 11).

Looking at the stresses, all the values are comparable except the minimum indicator when circulating from slab to ballast, as the minimums are very similar. The rest of the values are lower in the Dinatrans solution, meaning there was a slight reduction in stresses in the Dinatrans solution Table 10).

For vertical rail displacements (δ_rail_), only the maximum values are comparable when the train runs from ballast track to slab track. There the indicator indicates that the Dinatrans solution gave lower absolute displacements, approximately 15.32% lower (Table 11).

Finally, almost all the values for the relative displacement variable were comparable as there are significant differences between them, except for the maximum relative displacement corresponding to the direction of movement from ballast to slab track. The relative displacements were clearly lower in the Dinatrans solution.

There was a 14.12% reduction in vertical relative displacement with the Dinatrans solution compared with the Standard solution. Furthermore, there was a reduction of 3.70% (positive shear stress) and 1.49% of negative shear stress with the Dinatrans over the common solution (Table 11).

As can be seen, there are variables for which comparisons can be made for the maximum and not for the minimum and vice versa. This is because the differences between one and the other solution are small. In a comparative analysis and its statistical justification, it was found that the comparable values were improved with the Dinatrans solution. The rest of the values cannot be compared, because their differences were not significant enough to establish a comparison. The results for the stresses and the relative displacements were positive for the Dinatrans solution in the sense that they were reduced with respect to the Standard solution. However, it is difficult to come to firm conclusions with only these initial measurements and initial modeling. Therefore, long term measurements need to be taken to observe the evolution and the future behavior of both track transition solutions in similar environments.

Up until now (from 2016 to 2021), the authors have not seen any records of any maintenance work being done on either of the track transition areas, meaning no maintenance work has been necessary to date. The line has been used from 2018 to the present.

A long-term analysis is required to provide a complete understanding of the track transition performance. Some authors [46] have modeled long-term ballast vertical deflection in track transitions and further studies must continue along this path. Variables, such as sleeper spacing, have been found that are crucial for good track transition behavior [27,47].

The methodology used was to provide an initial track response before opening to traffic and provide an initial approach to assess the transient performance for both track transition solutions.

## 5. Conclusions

This document presents a description and the main results for an initial track test for two track transition structures. One of these is described as the common solution normally used in these areas (“Standard solution”) and the other, the proposed Dinatrans solution. Initial measurements were taken to compare the Dinatrans solution with the common solution.

Despite the fact that all measurements were taken using the same reference points in both places (slab track from the tunnel), and that both solutions were constructed in a similar way (e.g., same materials, same structure, same track geometry characteristics, ballast thickness, cant, etc.), this initial comparison was not straightforward, so statistical tools were used to enable initial comparison between these zones. This research has provided an initial approach which can be used to make comparisons with future measurements taken in both areas.

Both track transitions were within safety structural limits. The model values were found to be similar to the real measurements, except in the case of vertical displacements. Hanging sleepers were not considered in the initial numerical model. The influence of train travel direction was small for the variables considered in the analysis.

Statistical analysis was used to allow comparisons to be made between the two solutions (initially with similar results). While both track transitions showed similar performances at the beginning, the Dinatrans solution achieved some advantages over the common solution.

The Dinatrans solution is simple and easy to build and install on site and its installation cost is lower than that of the Standard solution in these areas. This new solution can use recycled materials, such as recycled rails, for the inner or outer rails used in the Dinatrans solution.

The Dinatrans solution was patented in (2018) [7]. Initial measurements showed slight improvements over the common solution, but this requires to be checked over the long term to provide a complete understanding of any performance advantage. The next steps consist of checking the solution in the medium- and the long-terms using this track site as a living railway track laboratory. Therefore, long-term models are necessary to check these solutions and predict their future behavior under operation. The Spanish railway administrator will continue making measurements of both these track transition solutions to determine long-term track performance and assess how they evolve.

## 6. Patents

There is a patent related to part of this work; ES 2684429 B1 (2018) Zona de transición de una línea ferroviaria situada entre una vía en balasto y una vía en placa de hormigón. Tomo 2 de Patentes y modelos de utilidad. de 09 de Julio de 2019. International patent classification: E01B 1/00 (2006.01)

## Figures and Tables

**Figure 1 sensors-22-00076-f001:**
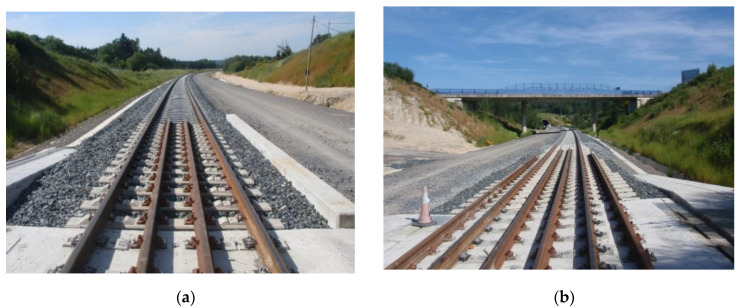
(**a**) Common solution for track transition used in Spain. (**b**) Dinatrans track transition.

**Figure 2 sensors-22-00076-f002:**
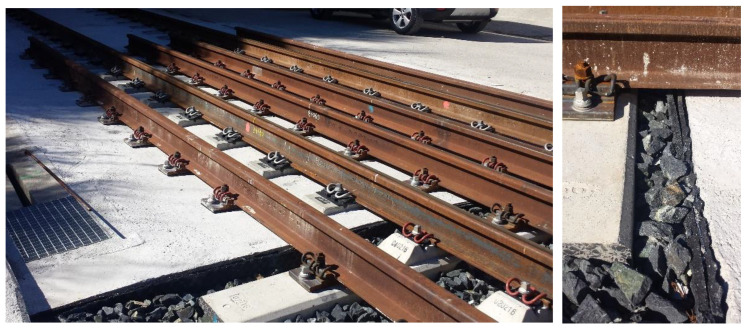
Detail of cross-section of the ballast mat between slab and ballast track in Dinatrans solution.

**Figure 3 sensors-22-00076-f003:**
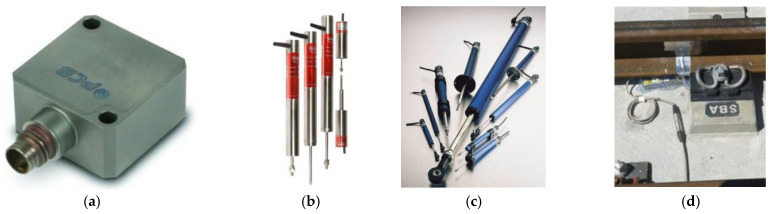
Accelerometers (**a**)**,** LVDTs (**b**), potentiometers (**c**) and extensometer gauges (**d**).

**Figure 4 sensors-22-00076-f004:**
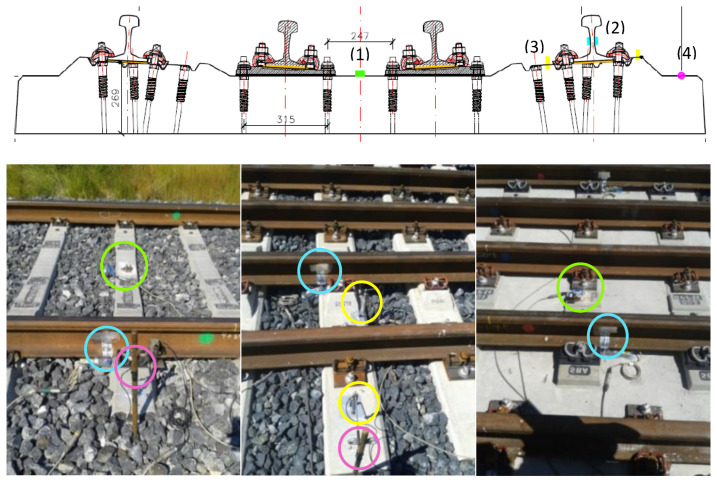
Schematic and location of the track measurement devices on the track. Position on the sleeper (**upper**). Examples of several monitored sleepers (**lower**).

**Figure 5 sensors-22-00076-f005:**
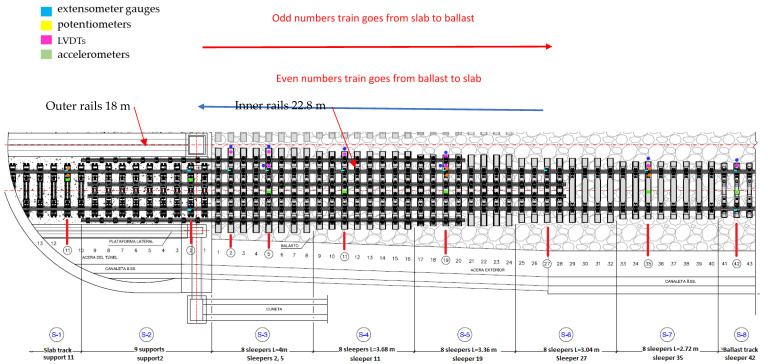
Dinatrans track transition. Location of the measuring sensors installed on the track. The same reference points were used in the common track transition solution.

**Figure 6 sensors-22-00076-f006:**
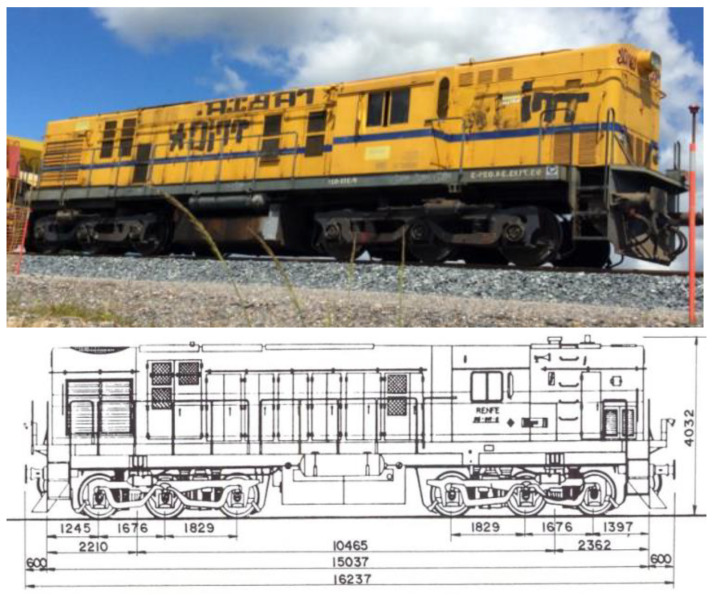
ALCO Locomotive Model 313, used in the model and field tests.

**Figure 7 sensors-22-00076-f007:**
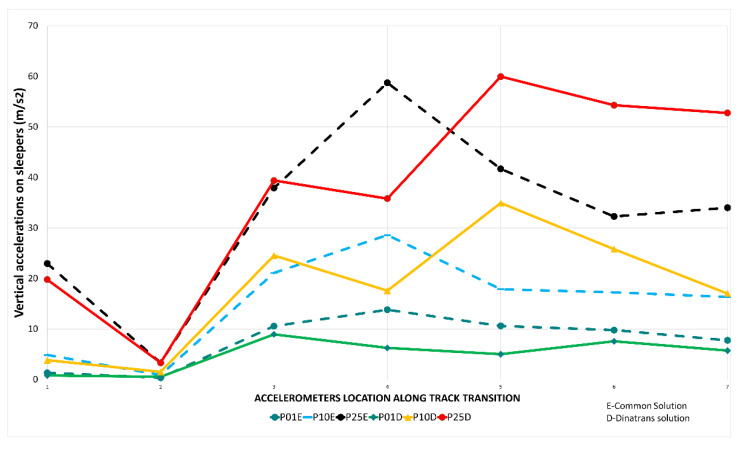
Vertical accelerations (m/s^2^) for sleeper vs. accelerometer position and passing trains in both track transitions. Dinatrans solution (D)-common solution (E).

**Figure 8 sensors-22-00076-f008:**
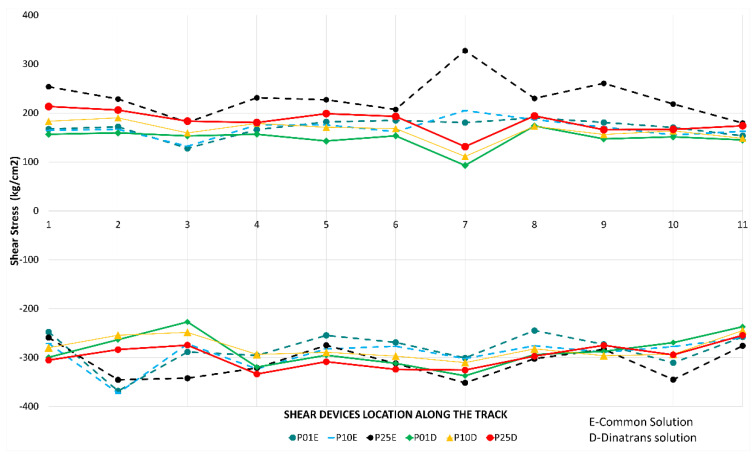
Shear stresses (MPa) vs. device location along the track in rail web and passing trains for both track transitions. Dinatrans solution (D)-common solution (E).

**Figure 9 sensors-22-00076-f009:**
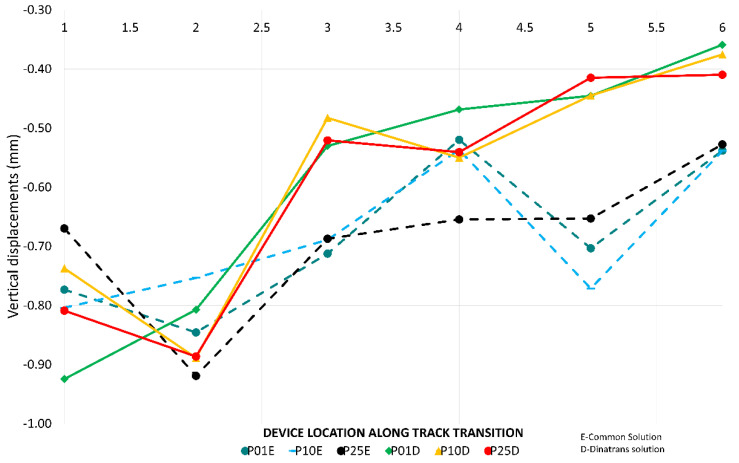
Vertical displacements between rail and sleeper (mm) versus device location along the transition and passing trains.

**Figure 10 sensors-22-00076-f010:**
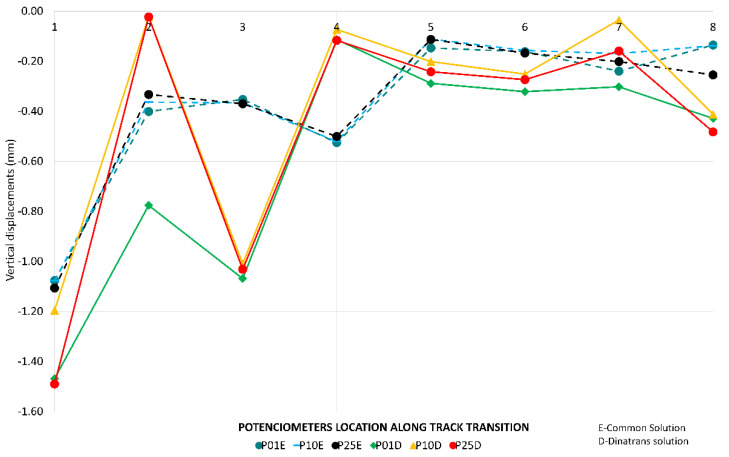
Vertical displacements on sleepers and supports.

**Figure 11 sensors-22-00076-f011:**
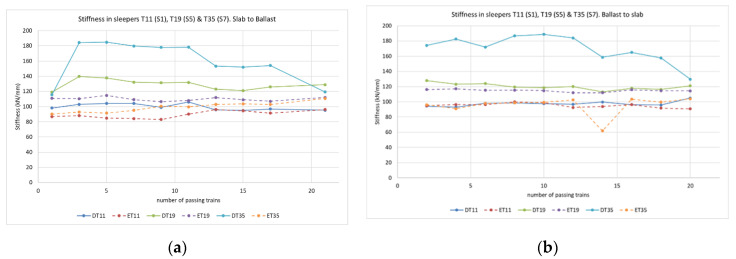
Stiffness in sleepers 11, 19 and 35. Train goes from slab to ballast (**a**). Train goes from ballast to slab (**b**).

**Figure 12 sensors-22-00076-f012:**
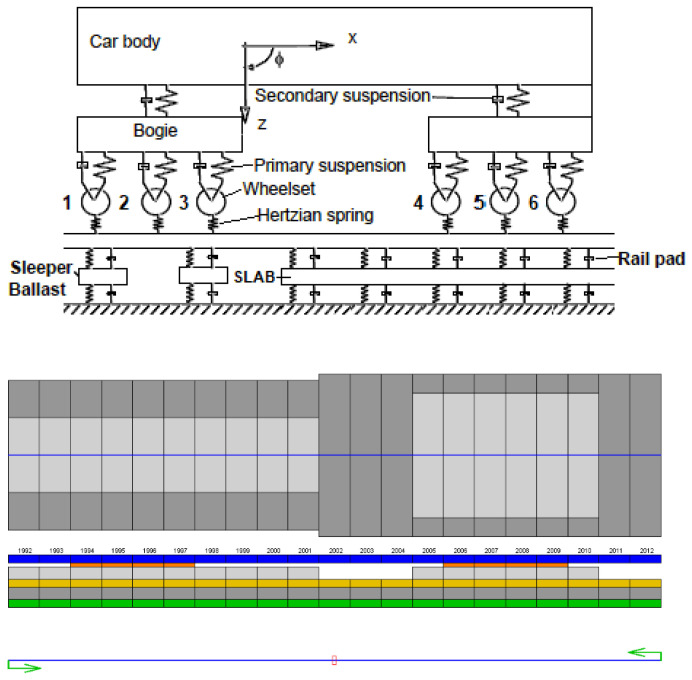
Scheme of the 2D model used in the simulation (**upper**)**.** Numerical model and element size of 0.05 m used in the simulation (**lower**).

**Figure 13 sensors-22-00076-f013:**
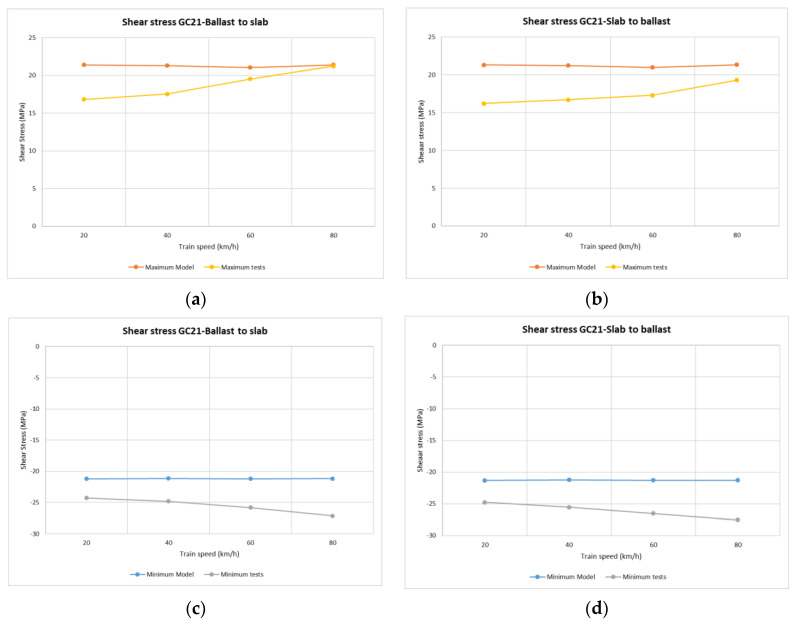
Shear stress from the Dinatrans model and in situ measurements. Depending on train speed and travel direction (support T2 section 1-slab track). (**a**,**b**) shows the tensile stress in rail web over support T2 (section 2-slab track). (**c**,**d**) shows compressive stress in rail web over support T2 (section 2-slab track). It can be seen that there are tensile and compressive stress in the rail web.

**Figure 14 sensors-22-00076-f014:**
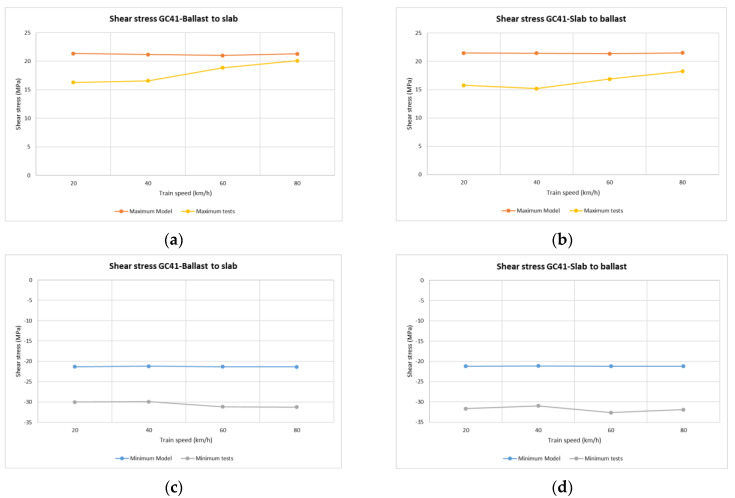
Shear stress from the Dinatrans model and in situ measurements. Depending on train speed and travel direction (sleeper T11 section 4-ballast track). (**a**,**b**) tensile stress in rail web over sleeper T11 (section 4-ballasted track). (**c**,**d**) compressive stress in rail web over sleeper T11 (section 4-ballasted track).

**Figure 15 sensors-22-00076-f015:**
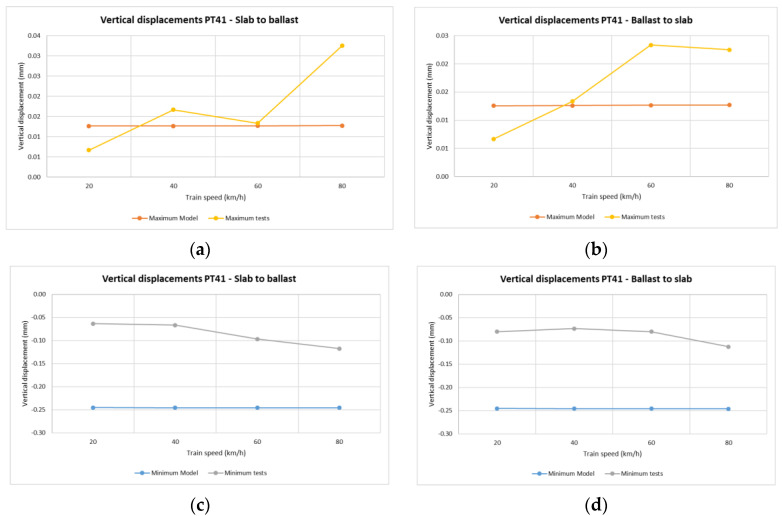
Vertical displacements of sleeper depending on train speed and travel direction. Dinatrans model and real measurements on the track (sleeper T11 section 4-ballasted track). Depending on train speed and travel direction (sleeper 11 section 4-ballast track). (**a**,**b**) Positive vertical displacements of sleeper T11 (section 4-ballasted track). (**c**,**d**) Negative vertical displacements of sleeper T11 (section 4-ballasted track).See Table 4.

**Figure 16 sensors-22-00076-f016:**
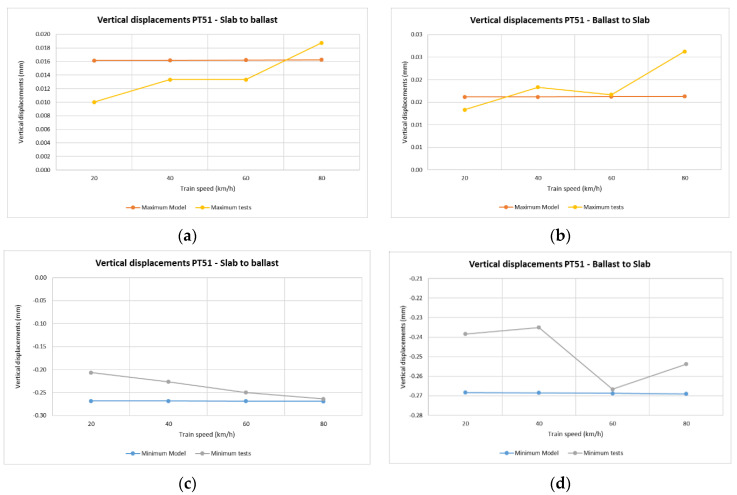
Vertical displacements of sleeper depending on train speed and travel direction. Dinatrans model and real measurements (sleeper T19 section 5-ballasted track). Depending on train speed and travel direction (sleeper T19 section 5-ballast track). (**a**,**b**) Positive vertical displacements of sleeper T19 (section 5-ballasted track). (**c**,**d**) Negative vertical displacements of sleeper T19 (section 5-ballasted track). See Table 4.

**Figure 17 sensors-22-00076-f017:**
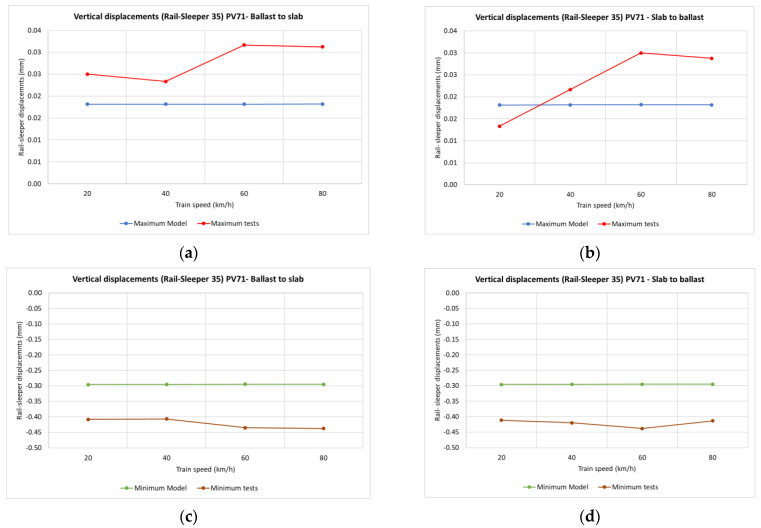
Vertical displacement of rail (between rail and sleeper 35 section 7-ballasted track). Depending on train speed and travel direction (sleeper T35 section 7-ballast track). (**a**,**b**) Positive vertical displacements between rail and sleeper T35 (section 7-ballasted track). (**c**,**d**) Negative vertical displacements between rail and sleeper T35 (section 7-ballasted track). See Table 4.

**Table 1 sensors-22-00076-t001:** Classification of different types of solution.

Type of Solution	Description
1. Acting on superstructure (Permanent way)	Add inner rails between both areas
	Add outer rails between different areas
	Pads with different stiffness
	Longer sleepers on ballast side (constant length)
	Sleepers with variable length along the track transition
	Using wood sleepers/rubber sleepers
	Under sleeper pads (USP)
	Glued ballast
	etc.
2. Acting on infrastructure (Formation)	Using geotextile layers under ballast
	Using geogrids between layers
	Hot mix asphalt layers (HMA)
	Transition slab track approach
	Transition wedges
	Concrete slab approaches
	etc.
3. Acting on subsoil (Natural ground)	Using Stone columns
	Pile foundation
	Slab foundation
	etc.
4. Mixed solutions	4.1 Combination of solutions from 1
	4.2 Combination of solutions from 2
	4.3 Combination of solutions from 3
	4.4 Combination between 1–2
	4.5 Combination between 2–3
	4.6 Combination between 1–3
5. Acting on rolling stock	Modifying spring and damping systems.

**Table 2 sensors-22-00076-t002:** Geotechnical properties of the ground (average values from field test initial and final pK).

PK Initial-PK Final	E (MPa)	u	ρ (t/m^3^)	C (MPa)	(j) (°)
3+440–4+930	485.1	0.27	2.55	0.196	50.05

**Table 3 sensors-22-00076-t003:** Main characteristics of sensors used in the field instrumentation.

Sensor	Sensitivity Range	Measurement Range	Resolution	Temperature Range	Frequency Range
Vertical Accelerometer	(±5%) 1000 mV/g (101.9 mV/(m/s^2^))	±2 g pk (±19.6 m/s^2^ pk)	0.25 mg rms (0.0025 m/s^2^ rms)	−65 to +250 °F (−54.0 to +121 °C)	(±5%) 0 to 250 Hz
LVDTs	±0.1%FS	±0.25 mm until ±470 mm	infinite	−58 °F a 257 °F (−50 °C a 125 °C)	>100 Hz
Potentiometer	High accuracy	Until 1600 mm	±0.2%	Maximum 350 °C	-
Extensometer gauge	Transversal sensitivity less than 0.3%	Linear gauges 0.3 mm to 20 mm Grids XY Gauges (Tee Rosettes) 0.6 to 6 mm Grids	-	<0.01%/°C. 10.8 × 10^−6^/°C	-

**Table 4 sensors-22-00076-t004:** Location of accelerometers, extensometer gauges, LVDTs and potentiometers to measure the variables during the field test.

Type of Track	Slab Track		Ballast Track						
Section	S1	S2	S3	S3	S4	S5	S6	S7	S8
Accelerometer (Device n°) x	AT11(1)	AT21(2)	AT31(3)		AT41(4)	AT51(5)		AT71(6)	AT81(7)
Sleeper number	T-11	T-2	T-5		T-11	T-19		T-35	T-42
Shear stress (Device n°) x	GC11(1)	GC21(2)	GC31(3)	GC32(4)	GC41(5)	GC51(6)	GC61(7)	GC71(8)	GC81(9)
Sleeper number	T-11	T-2	T-2	T-5	T-11	T-19	T-27	T-35	T-42
Rail-sleeper vertical displacement (Device n°) x	PV11(1)					PV51(2)		PV71(3)	
Sleeper number	T-11					T-19		T-35	
Sleeper vertical displacement (Device n°) x			PT31(1)	PT32(2)	PT41(3)	PT51(4)		PT71(5)	PT81(6)
Sleeper number			T-2	T-5	T-11	T-19		T-35	T-42

**Table 5 sensors-22-00076-t005:** Passing trains and velocity steps. Even numbers: passing trains travel from ballast to slab; odd numbers: train travels in the opposite direction.

Transition Type	Passing Trains	Velocity Steps	Speed
STANDARD SOLUTION (E)	1–6	V1	20
	7–12	V2	40
	13–18	V3	60
	19–26	V4	80
DINATRANS (D)	1–6	V1	20
	7–12	V2	40
	13–18	V3	60
	19–26	V4	80

**Table 6 sensors-22-00076-t006:** Geometry of the track model.

		Ballast Track	Slab Track
Rail		UIC 60	
Sleepers	Length (m)	2.6 m	-
	Height (m)	0.22	-
Slab	Length (m)	-	2.4
	Height (m)	-	0.246
Formation	Cross section (m)	5.2	4.8
	Thickness (m)	0.6	0.75
Ballast	Thickness (m)	0.3	-

**Table 7 sensors-22-00076-t007:** Mechanical properties of the track elements being considered.

		Ballast Track	Slab Track
Rail	UIC 60		
PADS (per unit of length)	K1 (kN/m)	500,000	165,000
	C1 (kN·s/m)	35	40
Sleepers	E (kN/m)	80,000,000	-
	Poisson Coef.	0.2	-
Slab	E (kN/m)	-	35,000,000
	Poisson Coef.	-	0.2
Ballast/elastic bed (per area unit)	K2 (kN/m)	200,000	80
	C2 (kN·s/m)	24,000,000	200
Foundation layer	K (kN/m)	1,000,000	
	C (kN·s/m)	20	

**Table 8 sensors-22-00076-t008:** Mechanical characteristics of the train.

Bogie	K1 (kN/m)	400
	C1 (KN·s/m)	50
Car body	K2 (kN/m)	300
	C2 (KN·s/m)	24
Total weight	(kN)	839

**Table 9 sensors-22-00076-t009:** Average of maximum values in all supports for each variable in both directions for Dinatrans and common solutions. DIN-Dinatrans values, EST-common solution values.

Variables	Travel Direction from Slab to Ballast			
	Maximum (Average)		Minimum (Average)	
	${\bar{x}}_{DMAX}\mathrm{DIN}$	${\bar{x}}_{EMAX}\mathrm{EST}$	${\bar{x}}_{DMIN}\mathrm{DIN}$	${\bar{x}}_{EMIN}\mathrm{EST}$
a (cg)	199.829	190.634	−225.343	−191.219
σ (kg/cm^2^)	163.792	189.148	−293.781	−291.329
δ_sleep_ (mm)	0.032	0.034	−0.446	−0.377
δ_rail_ (mm)	0.040	0.053	−0.601	−0.700
**Variables**	**Travel Direction from Ballast to Slab**			
	**Maximum (Average)**		**Minimum (Average)**	
	${\bar{x}}_{DMAX}$ **DIN**	${\bar{x}}_{EMAX}$ **EST**	${\bar{x}}_{DMIN}$ **DIN**	${\bar{x}}_{EMIN}$ **EST**
a (cg)	195.729	170.120	−201.982	−164.815
σ (kg/cm^2^)	170.881	177.448	−285.624	−289.940
δ_sleep_ (mm)	0.037	0.044	−0.409	−0.376
δ_rail_ (mm)	0.047	0.048	−0.591	−0.677

**Table 10 sensors-22-00076-t010:** Comparison indicators for the means of the maximum values between one solution and another for each direction of circulation.

Variable Indicators	Train Travel Direction from Slab to Ballast			
	Maximum		Minimum	
Indicators	*I* _max_	T-Student	*I* _min_	T-Student
a (cg)	104.822	0.352	117.845	−1.181
σ (kg/cm^2^)	86.594	−7.317	100.841	−0.579
δ_sleep_ (mm)	95.226	−0.353	118.275	−1.137
δ_rail_ (mm)	76.194	−2.440	85.879	3.241
	**Train Travel Direction from Ballast to Slab**			
	**Maximum**		**Minimum**	
**Indicators**	** *I* _max_ **	**T-Student**	** *I* _min_ **	**T-Student**
a (cg)	115.053	1.009	122.550	−1.440
σ (kg/cm^2^)	96.299	−2.232	98.511	1.239
δ_sleep_ (mm)	84.680	−1.259	108.859	−0.568
δ_rail_ (mm)	97.634	−0.240	87.335	2.977

**Table 11 sensors-22-00076-t011:** Percentages of variable improvement using the Dinatrans solution compared to the common solution.

% Dinatrans Improvement over Common Solution	Dinatrans Solution			
	from Slab to Ballast		from Ballast to Slab	
Indicators	Maximum%	Minimum%	Maximum%	Minimum%
a (cg)	4.82	17.85	15.05	22.55
σ (kg/cm^2^)	−13.41	0.84	−3.70	−1.49
δ_sleep_ (mm)	−23.81	−14.12	−2.37	−12.66
δ_rail_ (mm)	−4.77	18.28	−15.32	8.86

## Data Availability

No new data were created or analyzed in this study. Data sharing is not applicable to this article.

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
