# Peer review of "Monitoring Track Transition Zones in Railways"

_sensors, 2021, doi:10.3390/s22010076_

Round 1
Reviewer 1 Report
The paper compares the performance of transition zones that are provided with a Standard solution and the Dinatrans Solution. The study is of practical significance but the authors have not presented the key parameters like the density and thickness of ballast, positioning of inner and outer rails (in case of Dinatrans solution) and the justification of the length over which these additional rails are to be laid etc. The values of the various parameters presented by the authors seem to be too close in both the solutions, and hence it cannot be established with certainty the superiority of the solution proposed by the authors. As several important aspects are not properly described in the manuscript, the reviewer is of the opinion that the paper in its current form is not publishable. The specific comments are as follows;
The density and thickness of ballast should be mentioned in the manuscript. The positioning of inner and outer rails (in case of Dinatrans solution) and the justification of the length over which these additional rails are to be laid are to be described in the manuscript. What should be section of rail to be used for the same?
Will the length of inner and outer rails depend on the thickness of ballast or the stiffness of ballast layer? This needs to be clarified in the manuscript.
It is not clear when exactly (time: year) the track was constructed, monitored? How long the monitoring was done? Is the track monitored continuously with time and the passage of trains? Whether the track was opened for traffic?
Field measurements from track opened to traffic (comprising of both passenger and/or freight trains) will highlight the actual performance of transition. The single locomotive moving at relatively lower speed chosen in this study is not sufficient to capture the actual behaviour of transition. Additional data related to the performance of transition zone after the track is opened for the real traffic needs to be incorporated.
Line 126 to 129: Provide reference or justification for choosing different stiffness rail pads at the transition zone;
Line 146: The polystyrene cross section panel used at transition needs to be described and the photograph of the same should be shown in the manuscript.
Table 2: Elaborate the notations used in table.
Line 168 -171: Cite the references and also highlight the reasons for choosing the selected parameters in analyzing the transition performance.
Line 174: Figure 4 is cited before Figure 3 in text.
Lines 216-220 and Lines 221-225 are the same.
Line 257: Cite the reference.
Table 3: The notations S1 to S4 mentioned in the Table are also used for representing track sections in Figure 4. This needs to be corrected.
Line 313: How the transition zone is simulated? Suitable figure of numerical model highlighting the transition zone should be included in the manuscript.
Figure 10: Why do the shear stresses remains constant with increasing with speed?
Figure 11: The test results and modelling results does not match well.
Figure 12: ‘0.01 mm’ vertical displacement is an insignificant value to compare. Why does the settlement remain constant with train speed? What is the accuracy of the instrumentation used for recording the displacements? What is the repeatability of the observed settlement values?
The paper needs to be checked for grammatical mistakes.
Author Response
Please find attached the response to reviewer 1

Reviewer 2 Report
The manuscript "Monitoring track transition zones in railways" presents field measurements of a track transition zone performance in Galicia Spain. The obtained field measurements are then compared to the results of a numerical model.
Some points for improvement are provided bellow:
The literature review could be improved by including a significant number of papers reporting track transition performance measurements spanning the 2017-2021 time period.
The manuscript may be improved by minor typo editing.
It would be interesting to report on the relative stiffness of the soil relative to the stress cell. Also it would be useful to provide a more detailed Fig. of the in situ instrumentation.
Author Response
Please find attached the response to reviewer 2.

Round 2
Reviewer 1 Report
The paper has been revised suitably and is publishable in its current form.